# Influence of Operating Conditions on Reuterin Production Using Resting Cells of *Limosilactobacillus reuteri* DPC16

Yuanze Sun [1,*] , Noemi Gutierrez-Maddox [1], Anthony N. Mutukumira [2], Ian S. Maddox [2] and Quan Shu [3]

1 School of Science, Auckland University of Technology, 55 Wellesley Street East, Auckland CBD, Auckland 1010, New Zealand; iannoemi@xtra.co.nz
2 School of Food and Advanced Technology, Massey University, Auckland 0632, New Zealand; a.n.mutukumira@massey.ac.nz (A.N.M.); i.s.maddox@massey.ac.nz (I.S.M.)
3 Bioactive Research New Zealand, Auckland 1023, New Zealand; marsh.shu@gmail.com
* Correspondence: arthur.sun@autuni.ac.nz

**Abstract:** *Limosilactobacillus reuteri* strains can secrete a potentially valuable chemical and broad-spectrum antimicrobial substance named reuterin (3-hydroxypropionaldehyde, 3-HPA). *L. reuteri* DPC16 is a novel and patented probiotic strain that is used commercially because of its proven ability to kill various foodborne pathogens. A two-step process has been developed for reuterin production from glycerol using *L. reuteri* DPC16. Cells were grown, followed by harvesting, and then were incubated with glycerol for reuterin production. Parameters investigated during the glycerol conversion included the initial glycerol concentration, the biomass concentration, pH, culture age at harvesting, conversion time, and temperature. The highest reuterin yield was obtained using 21 g/L 24 h old cells, to convert glycerol solution (300 mmol/L) in 1 h at 30 °C and pH 6.2. The most efficient transformation of glycerol to reuterin was achieved in approximately 20 h of growth of cells at 25 °C and pH 6.8. Using the regression equation of this study, the maximum concentration of reuterin can be obtained using 25 g/L 20 h old DPC6 cells to ferment 350 mmol/L glycerol (initial concentration) for 2 h at 25 °C and pH 6.8 The ranking of effects on reuterin production for the six single factors was glycerol concentration > pH > conversion time > biomass concentration > temperature > culture age.

**Keywords:** *Limosilactobacillus reuteri* DPC16; reuterin; 3-hydroxypropionaldehyde; bioconversion; probiotics

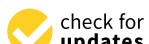



## 1. Introduction

The genus of the *Lactobacillus* species has been changed to *Limosilactobacillus*. In this new genus, 17 species have been identified. In the new taxonomy, *Lactobacillus reuteri* is now known as *Limosilactobacillus reuteri* [1]. *Limosilactobacillus reuteri* DPC16 (DPC16) was patented by Bioactives Research New Zealand Limited (previously named Dragon Pacific Limited) under the New Zealand Patents Acts 1953, after isolation from a healthy Caucasian male's faeces [2]. Comparison of the 16S rRNA gene fragment of DPC16 with GenBank data found that the strain was 99.3–99.6% similar to the known *Limosilactobacillus* genus and at least 0.5% of sequence difference compared to genomic DNA from other *L. reuteri* strains. It was shown that *L. reuteri* DPC16 was affiliated with the *Limosilactobacillus* genus [2,3]. The attractiveness of *L. reuteri* DPC16 as a probiotic organism is due to its antimicrobial effects against both Gram-positive and Gram-negative bacteria such as *Listeria monocytogenes*, *Staphylococcus aureus*, *Salmonella typhimurium* and *Escherichia coli* O157:H7 across a wide range of pH values and temperatures (as low as 10 °C) [3–5]. Further, *L. reuteri* DPC16 manifested good stability in simulated gastrointestinal fluids [6]. In vivo, there were over $10^4$ cells/g of wet human faeces after being administered as a single oral food sample containing $10^9$ organisms during a 10-day post-dosing period [2].

The antimicrobial activity of the supernatant of *L. reuteri* DPC16 was pH-independent over pH 4.6 to 6.5 [5]. The supernatant showed a bactericidal effect against the pathogens at high concentrations, while it showed a bacteriostatic effect at lower concentrations.

Reuterin (3-hydroxypionaldehyde, 3-HPA) is highly water-soluble, as it contains hydroxyl and aldehyde groups [7]. Three chemical forms, monomeric, hydrated monomeric and cyclic dimeric, can be formed when 3-HPA is in solution and presents a dynamic equilibrium. In aqueous solutions, 3-HPA can combine with water to form HPA hydrate to form acrolein through dehydration. The HPA dimer can be formed by combining two 3-HPA molecules. The three systems can mutually transform simultaneously and become an equilibrium mixture (Figure 1) [7].

**Figure 1.** The structure of reuterin [7].

Reuterin produced by *L. reuteri* strains is one of the reasons for the probiotic effect of pathogens [8]. Lactic and acetic acids are also produced by the strains and contribute to the antimicrobial effects of culture supernatants at low pH [5]. A significant amount of monomeric reuterin has never been detected. In aqueous solutions with high concentrations of reuterin, the main component of the HPA system is the HPA cyclic dimer, while the mole fraction of HPA hydrate increases as the reuterin concentration decreases [9].

The mechanism by which reuterin exerts its antimicrobial effect is reported to be difficult to determine [5]. The first hypothesis is that reuterin can form acrolein, which contains an aldehyde group that can react with free thiol groups. This reaction between the aldehyde group and glutathione (GSH) and the modification of proteins causes toxicity in microbial cells [10]. The second hypothesis is related to its dimer form. The HPA dimer can block DNA synthesis. The dimer works as a competitive inhibitor of the enzyme ribonucleotide reductase, as the structure of the HPA dimer is similar to that of a ribose sugar [11].

In vivo, *L. reuteri* strains produce reuterin enzymatically from glycerol using the enzyme glycerol dehydratase (GDHt) (Figure 2) [12]. Different sources of glycerol dehydratase have similar protein structures and similar molecular weights (188–196 kDa), but have different cation selectivities, coenzyme affinities and substrate specificities, thus leading to a different optimum pH (6.0–9.0) and temperature (28–37 °C) [12,13].

Glycerol dehydratase from *L. reuteri* strains has been shown to be co-enzyme Vb-12-dependent [12]. Combined actions occur between glycerol dehydratase and co-enzyme Vb12 to produce an electron donor that is used to convert glycerol to produce 3-HPA [13]. *L. reuteri* strains have been reported to have a strong glycerol conversion ability; however, the reuterin production yield is generally low. Reuterin production is known to be affected by oxygen concentration, glucose concentration and other environmental factors, such as temperature, pH, incubation time, cell age and biomass concentration. Glycerol dehydratase can be produced when *L. reuteri* strains are grown anaerobically on glycerol [14]. However, reuterin production is inhibited by the excess glucose present in glycerol solution because the NAD+ that is produced during conversion of glycerol to reuterin can be used in glucose metabolism. Reuterin can be further converted to 1,3-propanediol (1,3-PD) using propanediol dehydrogenase, depending on the NAD+ concentration. Thus, reuterin

production using *L. reuteri* has been reported to depend on the glucose: glycerol molar ratio. For maximum reuterin production, the ratio has been reported as 0.33 [15].

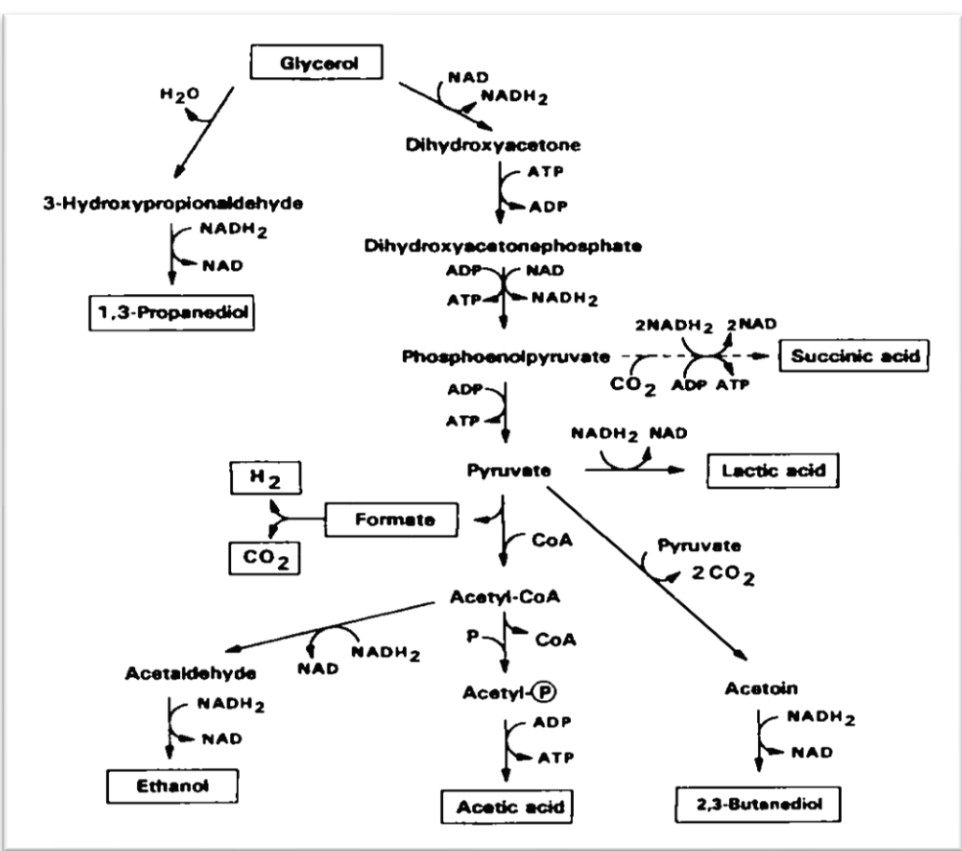

**Figure 2.** Glucose metabolic pathway in *L. reuteri* [13].

The chemical synthesis method of reuterin in the industry is associated with some disadvantages such as expensive catalysts, high reaction conditions (e.g., high temperature, high pressure) and environmental pollution [16,17]. Therefore, the biotechnology method used to produce reuterin may be cost-effective, and it works under milder production conditions. The biosynthesis of reuterin involves the conversion of glycerol using glycerol dehydratase as a catalyst to remove a $H_2O$ molecule of glycerol to generate reuterin in microbial cells. Glycerol is the termination substrate for all glycerol dehydratase enzymes, which indicates that glycerol dehydratase enzymes continuously decrease during the process, and the continuity of industrial production is thus affected by the inactivation of glycerol dehydratase enzymes. However, both *L. reuteri* cells and glycerol dehydratase can be reactivated. *L. reuteri* can be reactivated by aerobic incubation; meanwhile, glycerol dehydratase can also be reactivated in the presence of Vitamin B12 and ATP in a culture medium [18–21]. This mechanism enables the continuous production of reuterin using resting cells. In the current work, the purpose was to develop a two-stage process, using resting cells to maximise the conversion of glycerol to reuterin for potential commercial application. The study involved growing cells using a suitable growth medium, harvesting and washing them, followed by their suspension in glycerol solution for bioconversion. Individual factors of temperature, glycerol concentration, pH, biomass concentration, cell age at harvest and incubation time were investigated for their effects on reuterin production.

## 2. Material and Methods

### 2.1. Bacterial Strain and Media

*Limosilactobacillus reuteri* DPC16 was obtained from Drapac NZ Ltd. (Auckland, New Zealand). The growth media employed were MRS broth and MRS agar, obtained

from Fort Richard Laboratories Ltd. (Auckland, New Zealand). All media were sterilized at 120 °C for 15 min.

## 2.2. Reagents

Acrolein (purity 97%) solution was purchased from O2Si Smart Solutions Ltd. (7290-B Investment Drive, North Charleston, SC, USA). L-tryptophan powder (purity 99.5%), glycerol analysis standard solution (purity 98.5%), 1,3-propanediol standard solution (purity 98%) and D-(+)-glucose standard powder (purity 99.5%) were supplied by Sigma-Aldrich New Zealand Co Ltd. (Auckland, New Zealand), Toluene (99.5%, Merck Ltd., Darmstadt, Germany), and concentrated hydrochloric acid (36.5–38%, Merck Ltd., Auckland, New Zealand) were obtained from the AUT University laboratory.

## 2.3. Biomass Production

Frozen *L. reuteri* DPC16 cells were reactivated by transferring an inoculum (10%) into MRS broth and incubating it for 24 h at 37 °C under 5% $CO_2$. The culture was used as the seed culture after a second transfer.

## 2.4. Dry Cell Mass

*L. reuteri* DPC16 strain was cultured as previously described, and 13 mL samples were aseptically withdrawn every 4 h. A total of 3 mL of each cultured sample were used for measuring absorbance at 620 nm (Pharmacia Biotech Spectrophotometer, Ultrospec 2000, London, UK) in a 4.5 mL plastic cuvette with a width of 10 mm. Then, 10 mL of cultured sample were centrifuged at $3100 \times g$ for 10 min at 4 °C to collect fresh cells of the *L. reuteri* DPC16 strain. The fresh cells were dried at 80 °C to constant weight.

Dry cell weight and absorbance values showed a linear relationship (Figure 3). The regression equation for the yields of dry cell weight was $Y = 0.4893X - 0.2593$ ($Y$: dry DPC16 cell weight, $X$: absorbance values, $R^2 = 0.9981$).

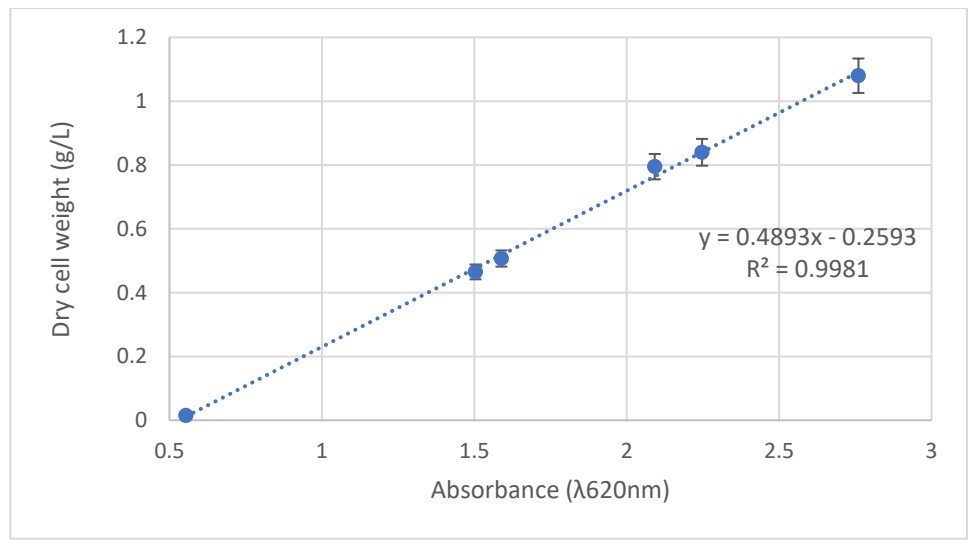

**Figure 3.** Harvested dry weight of *L. reuteri* DPC16 against the absorbance$_{620nm}$ of the 24-h old culture; $n = 3$; error bars = standard deviations of replications; analyses of experiments were triplicate.

## 2.5. Production of Reuterin from Glycerol

After growth in MRS broth, the bacterial cells were harvested by centrifugation (50 mL) at $3100 \times g$ for 10 min at 4 °C. The supernatant was discarded, and the sediment was washed twice with 1% peptone water before suspension in 1.3 mL of the appropriate conversion solution, contained in 10 mL centrifuge tubes.

The effects of culture age at harvest from the MRS growth medium (12, 16, 20, 24, 28, 32 h), glycerol concentration (150, 200, 300, 350, 400, 450 mmol/L), pH (adjusted using

0.2 M phosphate buffer to 6.0, 6.2, 6.8, 7.2, 7.5, 8), incubation temperature (20, 25, 30, 37, 42 °C), biomass concentration (9, 11, 17, 21, 25, 30 g dry cell weight/L) and incubation time (0.5, 1, 1.5, 2, 3, 4 h) were studied for 3-HPA production. After the conversion, cultures were centrifuged at 3100× *g* for 10 min at 4 °C. The supernatants were freeze stored at −80 °C until required for analysis.

*2.6. Analytical Methods*

The analysis of the acrolein concentration was based on the method of Circle et al. [22], with minor modifications. The standard L-tryptophan solution was prepared by dissolving 0.4017 g of L-tryptophan into 60 mL of concentrated HCl and adding 0.25 mL of toluene before increasing the volume to 100 mL. Briefly, 0.5 mL of sample was mixed with 1 mL of standard L-tryptophan solution and heated for 20 min at 40 °C to give it a purple color. The absorbance of the solution was measured at 570 nm in a Hitachi spectrophotometer (U-3900, Japan) (10–2-mm width fused quartz microcalorimetric cuvette).

Concentrations of reuterin, 1,3-propanediol and glycerol were determined using an Agilent Technologies HPLC (1200 series). Separation proceeded at 55 °C using an Aminex HPX-87H ion exclusion column (300 nm × 7.8 mm) (10 µL sample with 0.5 mL/min). The eluent solution was 5 mmol/L $H_2SO_4$, set at a flow rate of 0.5 mL/min. The concentrations of three compounds were measured using an external standard method and were recorded in the database.

Theoretically, glycerol is converted to reuterin (1 mole of glycerol is converted to 1 mole of reuterin) [9]. However, glycerol is also converted to other compounds, and the partially synthesized reuterin is further converted to 1,3-PD [23]. The glycerol consumption (%) and the reuterin yield (%) were calculated using Equations (1)–(3):

$$glycerol\ consumption\ (\%) = consumed\ glycerol\ (\mathrm{mmol/L}) \times 100/\ initial\ glycerol\ concentration\ (\mathrm{mmol/L}) \quad (1)$$

$$glycerol\ consumed\ (\mathrm{mmol/L}) = initial\ glycerol\ concentration\ (\mathrm{mmol/L}) - residual\ glycerol\ concentration\ (\mathrm{mmol/L}) \quad (2)$$

$$reuterin\ yield\ (\%) = reuterin\ concentration\ (\mathrm{mmol/L}) \times 100/\ initial\ glycerol\ concentration\ (\mathrm{mmol/L}) \quad (3)$$

*2.7. Statistical Analysis of Reuterin Production*

Data analysis on the inter-relationships of incubation time (h), age of cells (h), conversion temperature (°C), biomass concentration (g/L) and initial glycerol concentration (mmol/L) by the Principal Component Analysis (PCA) method was performed using IBM SPSS Statistics Version 21 (SPSS Inc., Chicago, IL, USA). Descriptive statistics of Microsoft Excel 365 (Office Inc., Washington, DC, USA) was used to analyze data on the response factors.

## 3. Results

*3.1. Effect of Biomass Concentration on Production of Reuterin*

The effect of biomass concentration on glycerol bioconversion was studied using DPC16 cells harvested after 24 h of growth. The cells were suspended in 300 mmol/L glycerol solution (pH 6.2) at 37 °C, and the conversion was measured after 1 h of incubation.

The amounts of glycerol consumed and reuterin produced increased significantly ($p < 0.05$) as the biomass concentration increased from 9 g/L to 21 g/L (Figure 4). The peak glycerol consumption (196.99 mmol/L) and reuterin production (180.76 mmol/L) occurred at 21 g/L biomass and were maintained until the biomass concentration had increased to 25 g/L. After the peak, glycerol consumption and reuterin production both decreased. Furthermore, there were very small amounts of 1,3-PD (6.80 ± 2.41 mmol/L) produced from reuterin, and its concentration was proportional to the biomass concentration.

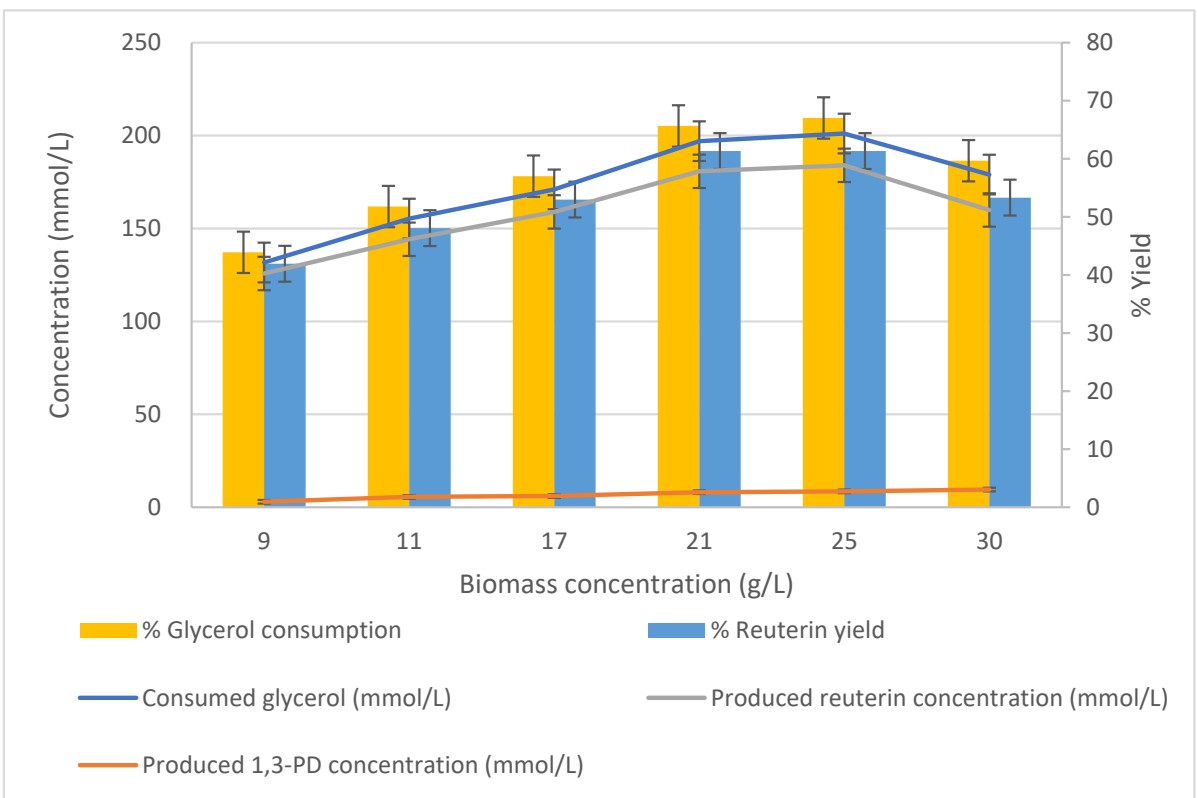

**Figure 4.** Effect of biomass concentration on reuterin production.

The effect of biomass concentration on glycerol bioconversion was studied using *L. reuteri* DPC16 cells harvested after 24 h of growth at 37 °C. The cells were suspended in 300 mmol/L glycerol solution (pH 6.2) at 37 °C and the conversion was measured after 1 h of incubation; *n* = 3; error bars = standard deviation of the means; analyses of experiments were triplicate

The amount of glycerol consumption (%) and the reuterin yield (%) followed similar trends. The peak glycerol consumption (67.0%) occurred at a biomass concentration of 25 g/L, while the peak reuterin yield (61.3%) occurred at biomass concentrations of 21 g/L and 25 g/L.

Under the experimental conditions used, the biomass concentration that maximized reuterin production was 21 g/L (Figure 4).

### 3.2. Effect of pH on Production of Reuterin

The effect of pH on the glycerol bioconversion was studied using cells harvested after 24 h of growth. DPC16 cells (25 g/L DW) were suspended in 300 mmol/L glycerol solution (37 °C), and the conversion was measured after 1 h of incubation. The pH of the glycerol solution was adjusted to 6.0, 6.2, 6.8, 7.2, 7.5 and 8.0 using 0.2 M phosphate buffer.

The relationship between different pH values and concentrations of glycerol, reuterin and 1,3-PD are shown in Figure 5. The peak glycerol consumption (204.61 mmol/L) occurred at pH 6.2. This consumption was significantly higher than that (175.89 mmol/L) at pH 6.0 ($p < 0.05$). Above pH 6.2, glycerol consumption decreased, although it remained relatively high. In contrast, the production of reuterin increased from pH 6.0 (153.60 mmol/L) to pH 7.2 (182.17 mmol/L), then decreased by a small amount. When the pH was above 7.5, which is the extreme pH for isolated glycerol dehydratase, the activity of intracellular glycerol dehydratase remained stable. The production of 1,3-PD peaked at pH 6.2, but the concentration was relatively low compared to the produced reuterin. Glycerol consumption showed a peak of 68.20% at pH 6.2. As the pH increased, the consumption of glycerol decreased. In contrast, as the pH increased, the yield of reuterin increased from 51.2%

(pH 6.0) to 60.7% (pH 7.2) before decreasing to 54.1% (pH 8.0). Thus, the optimum pH for producing reuterin using DPC16 was determined as 7.2 (Figure 5).

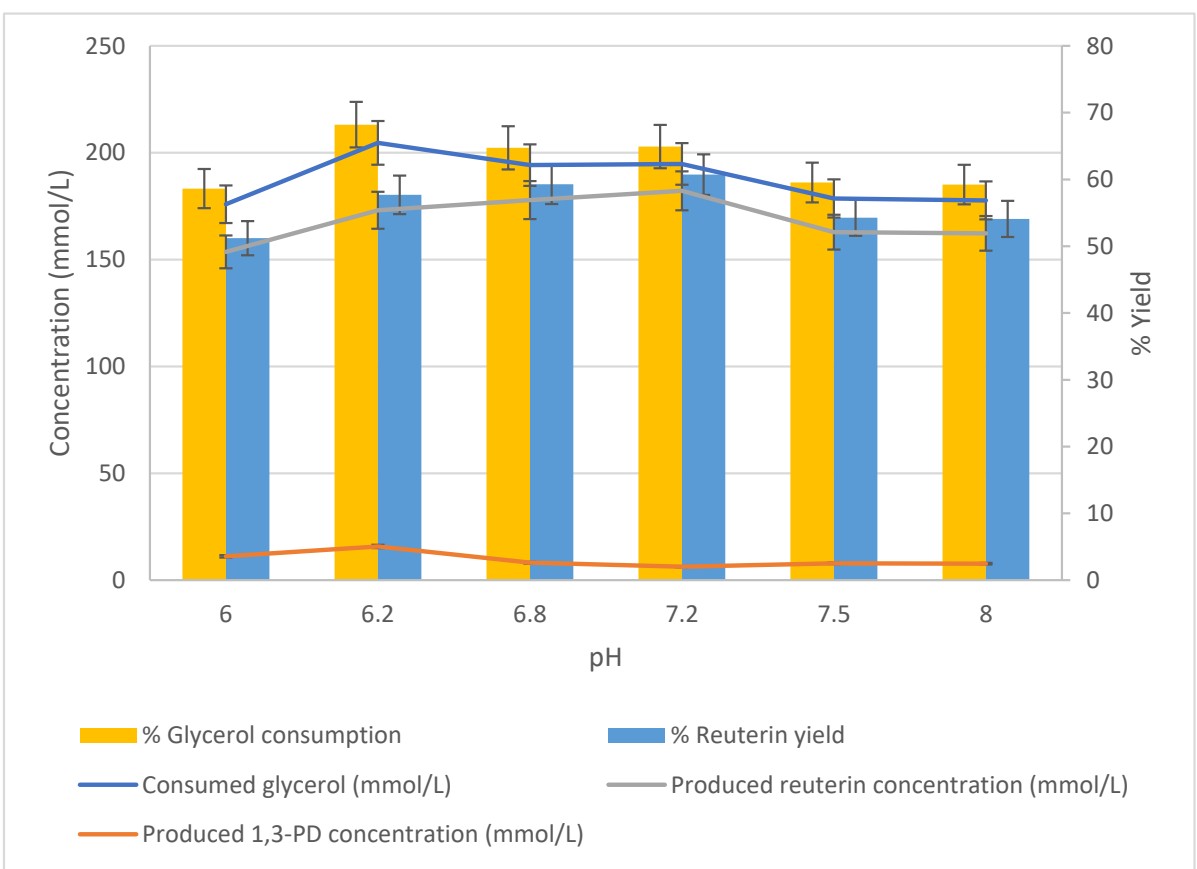

**Figure 5.** Effect of pH on reuterin production.

The effect of pH on the glycerol bioconversion was studied using *L. reuteri* DPC16 cells harvested after 24 h of growth. DPC16 cells (25 g/L DW) were suspended in 300 mmol/L glycerol solution (37 °C), and the conversion was measured after 1 h of incubation. The pH of the glycerol solutions was adjusted to 6.0, 6.2, 6.8, 7.2, 7.5, and 8.0 using 0.2 M phosphate buffer; $n = 3$; error bars = standard deviations of the means; analyses of experiments were triplicate.

### 3.3. Effect of Glycerol Concentration on Production of Reuterin

The effect of the initial concentration of glycerol during conversion to reuterin was studied using cells harvested after 24 h of growth. The cells (25 g/L) were suspended in a series of glycerol solutions (pH 6.2) at 37 °C, and the conversion was measured after 1 h of incubation. The concentrations of the glycerol solutions were 150, 200, 300, 350, 400 and 450 mmol/L.

The consumption of glycerol increased as the initial glycerol concentration increased from 150 mmol/L (86.35 mmol/L) to 350 mmol/L (230.38 mmol/L) (Figure 6). When the initial glycerol concentration exceeded 350 mmol/L, its consumption decreased. A similar trend was observed for reuterin production. The peak production of reuterin (212.74 mmol/L) occurred at 350 mmol/L of initial glycerol. Concentrations of produced 1,3-PD remained low throughout. When the initial glycerol concentration was 150 mmol/L, the supplied glycerol consumption was 57.6%. As the initial glycerol concentration increased from 200 mmol/L to 350 mmol/L, the glycerol consumption was approximately 65% of that supplied, after which point it decreased sharply. The relationship between reuterin yield and the initial glycerol concentration was similar to the relationship between the percentage

of glycerol consumed and the initial glycerol concentration. The reuterin yield was 45.5% at an initial glycerol concentration of 150 mmol/L; as the initial glycerol concentration increased, the reuterin yield also increased to a peak (60.8%) at 350 mmol/L of initial glycerol. After this point, the reuterin yield decreased as the initial glycerol concentration increased. Overall, the optimum initial glycerol concentration was 350 mmol/L for the conversion of glycerol to reuterin (Figure 6).

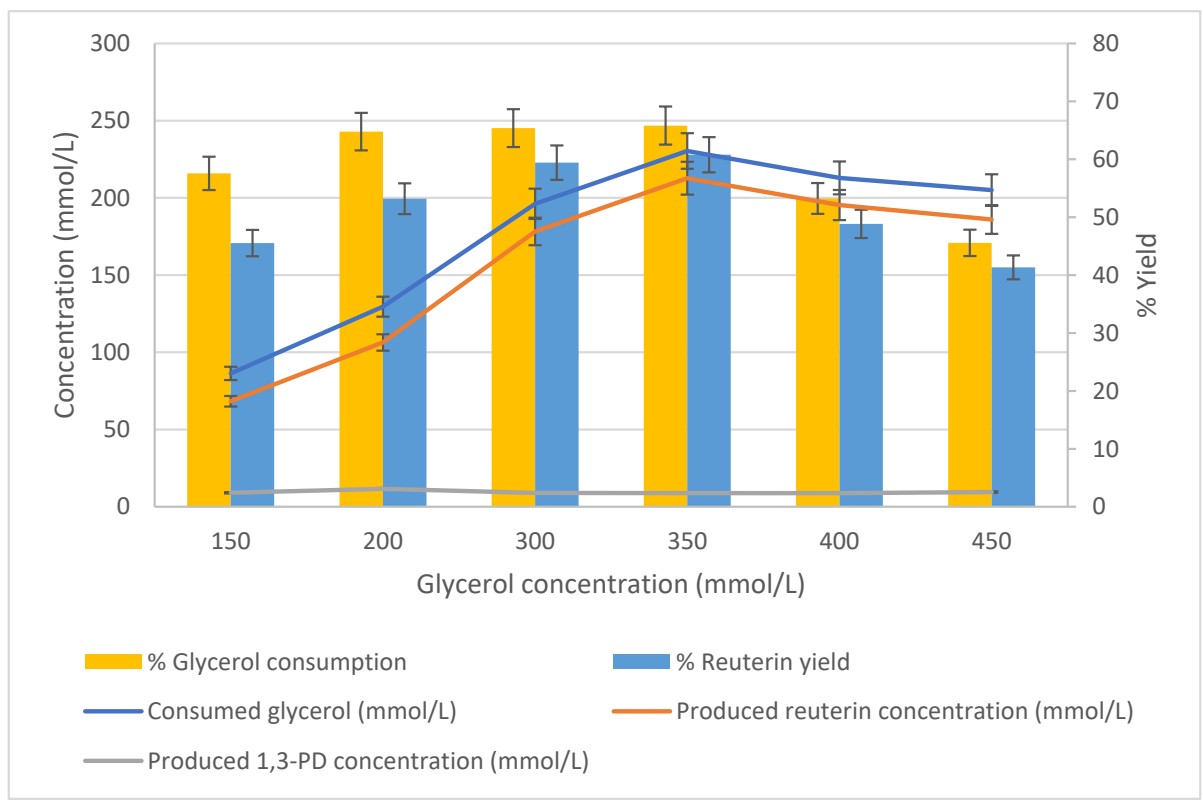

**Figure 6.** Effect of initial glycerol concentration on reuterin production.

The effect of the initial concentration of glycerol during conversion to reuterin was studied using cells harvested after 24 h of growth. The cells (25 g/L) were suspended in a series of glycerol solutions (pH 6.2) at 37 °C, and the conversion was measured after 1 h of incubation at 37 °C six concentrations of glycerol solutions: 150, 200, 300, 350, 400 and 450 mmol/L; *n* = 3; error bars = standard deviations of the means; analyses of experiments were triplicate.

### 3.4. Effect of Temperature on Production of Reuterin

The effect of temperature on glycerol bioconversion was studied using cells harvested at 24 h of growth. The cells (25 g/L) were suspended in 300 mmol/L glycerol solution (pH 6.2) and incubated at variable temperatures, with the conversion rate measured after 1 h of incubation. The tested temperatures were 20 °C, 25 °C, 30 °C, 37 °C and 42 °C.

At the two lower temperatures (20 °C, and 25 °C), the concentrations of consumed glycerol were similar, and the peak glycerol consumption (177.4 mmol/L) occurred at 25 °C (Figure 7). As the temperature increased, the consumption of glycerol decreased. The trend of reuterin production was similar to that of glycerol consumption, and the peak 177.40 mmol/L was observed at 25 °C. As the temperature increased, reuterin production decreased. The highest production of 1,3-PD was achieved at 20 °C, and then it stabilized.

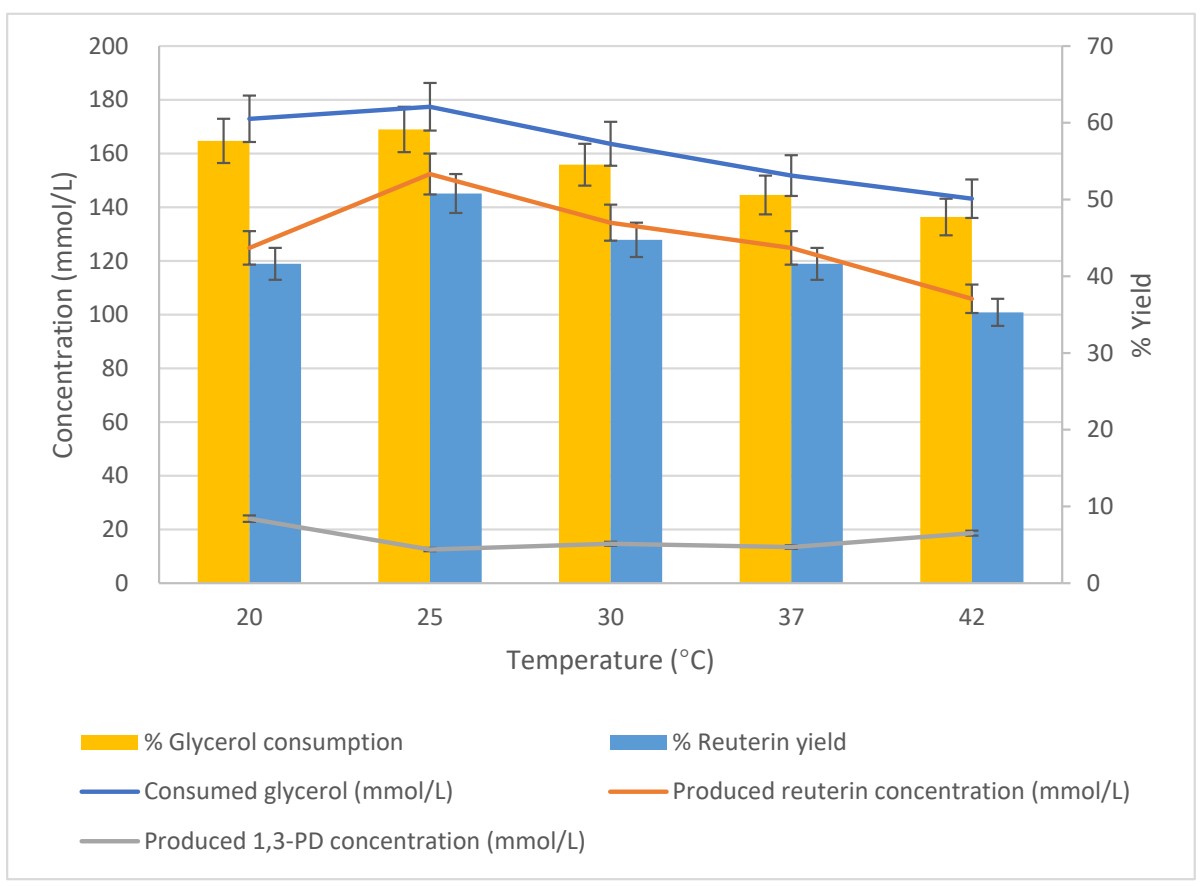

**Figure 7.** Effect of temperature on reuterin production.

The effect of temperature on glycerol bioconversion was studied using cells harvested at 24 h of growth. The cells (25 g/L) were suspended in 300 mmol/L glycerol solution (pH 6.2) and incubated at variable temperatures with the conversion rate measured after 1 h of incubation at four temperatures: 20 °C, 25 °C, 30 °C, 37 °C and 42 °C. *n* = 3; error bars = standard deviations of the means; analyses of experiments were triplicate.

Similarly, the consumption of glycerol was highest at 20 °C (57.6%) and 25 °C (59.3%). Above 25 °C, the reuterin production decreased as the temperature increased. However, the reuterin yield decreased sharply as the temperature changed. The reuterin yield was 41.6% at 20 °C, and the yield increased to a peak of 50.8% at 25 °C. As the temperature increased further, the reuterin yield decreased, reaching only 35.3% at 42 °C. Overall, the optimum temperature for the bioconversion of glycerol to reuterin by *L. reuteri* DPC16 was 25 °C, and the production concentration was 152.35 mmol/L (Figure 7).

*3.5. Effect of Biomass Concentration on Production of Reuterin*

The effect of incubation time on glycerol bioconversion was studied using cells harvested after 24 h of growth. The cells (25 g/L) were suspended at pH 6.2 and 37 °C in 300 mmol/L glycerol solution, and the glycerol-to-reuterin conversion was measured at a series of incubation times. The tested incubation times were 0.5, 1.0, 1.5, 2.0, 3.0 and 4.0 h.

Glycerol consumption increased as the incubation time increased to 2 h, after which it decreased (Figure 8). The effect of the conversion time on reuterin production followed a similar trend as that of glycerol consumption. The maximum consumption of glycerol (72.7%) was observed at 2 h of incubation. As the incubation time increased, the production of 1,3-PD increased throughout the entire period of incubation. The maximum consumption of glycerol and the reuterin yield were 72.7% and 59.4%, respectively. The maximum concentration of produced reuterin was 178.31 mmol/L (Figure 8).

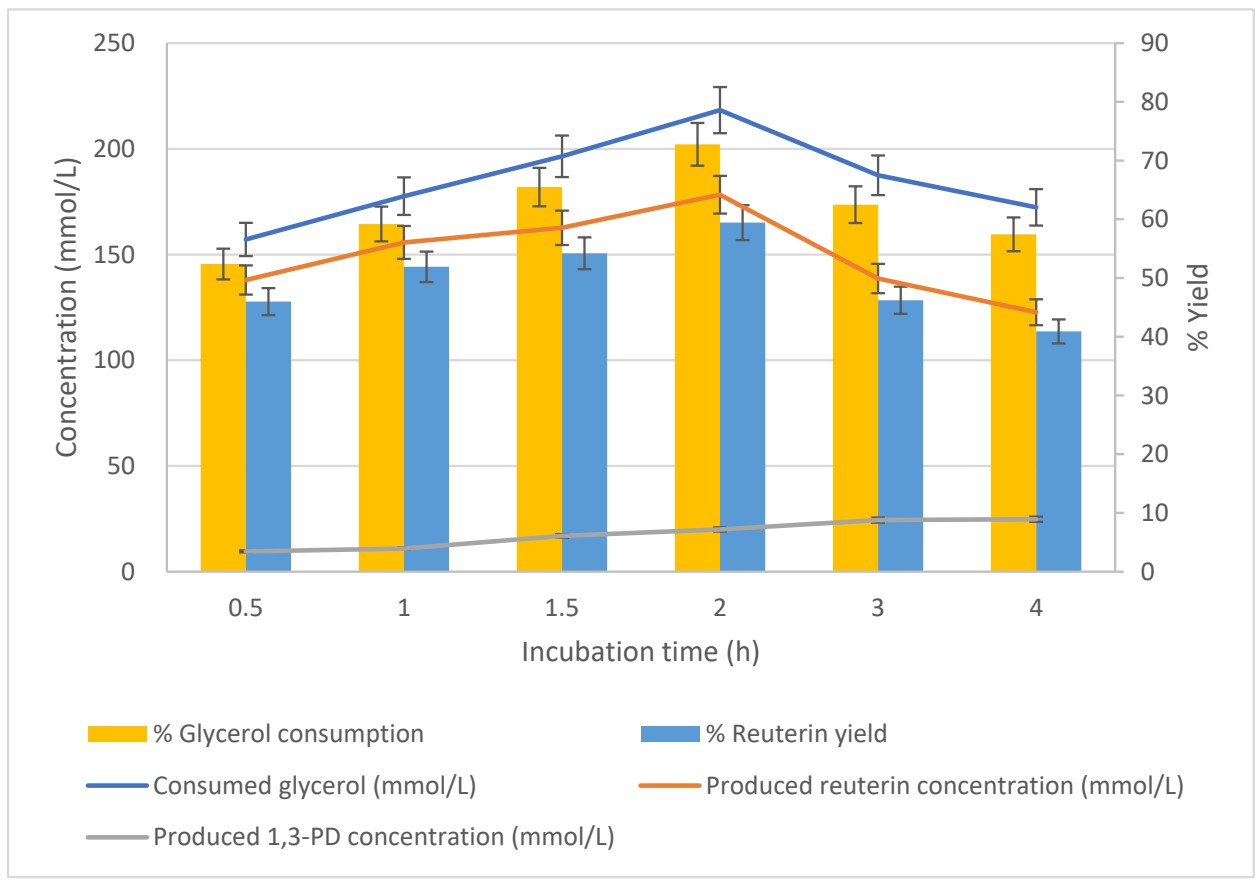

**Figure 8.** Effect of incubation time on reuterin production.

The effect of incubation time on glycerol bioconversion was studied using cells harvested after 24 h of growth. The cells (25 g/L) were suspended at pH 6.2 and 37 °C in 300 mmol/L glycerol solution, and the glycerol-to-reuterin conversion was measured during a series of incubation times. The tested incubation times were 0.5, 1.0, 1.5, 2.0, 3.0 and 4.0 h. $n = 3$; error bars = standard deviations of the means; analyses of experiments were triplicate.

### 3.6. Effect of Culture Age on Production of Reuterin

The effect of the culture age on glycerol bioconversion was studied using cells harvested after a series of different growth periods. The cells (25 g/L) were suspended at pH 6.2 and 37 °C in 300 mmol/L glycerol solution, and the conversion was measured after a 1 h incubation time. The tested culture ages were 12, 16, 20, 24, 28 and 32 h. Exponential growth had ceased after 16 h (data not shown).

The relationship between the culture age at harvest and the glycerol-to-reuterin conversion is shown in Figure 9. The concentrations of consumed glycerol (197.42 mmol/L) and reuterin production (176.07 mmol/L) both peaked at a harvest age of 20 h, although the levels did not vary considerably for cells harvested at different times.

In summary, the results showed the optimum culture age of DPC16 cells for harvesting cells for reuterin production was 20 h.

The effect of culture age on glycerol bioconversion was studied using cells harvested after a series of different growth periods. The cells (25 g/L) were suspended at pH 6.2 and 37 °C in 300 mmol/L glycerol solution, and the conversion was measured after 1 h incubation. The tested culture ages were 12, 16, 20, 24, 28 and 32 h. Exponential growth had ceased after 16 h (data not shown). $n = 3$; error bars = standard deviations of the means; analyses of experiments were triplicate.

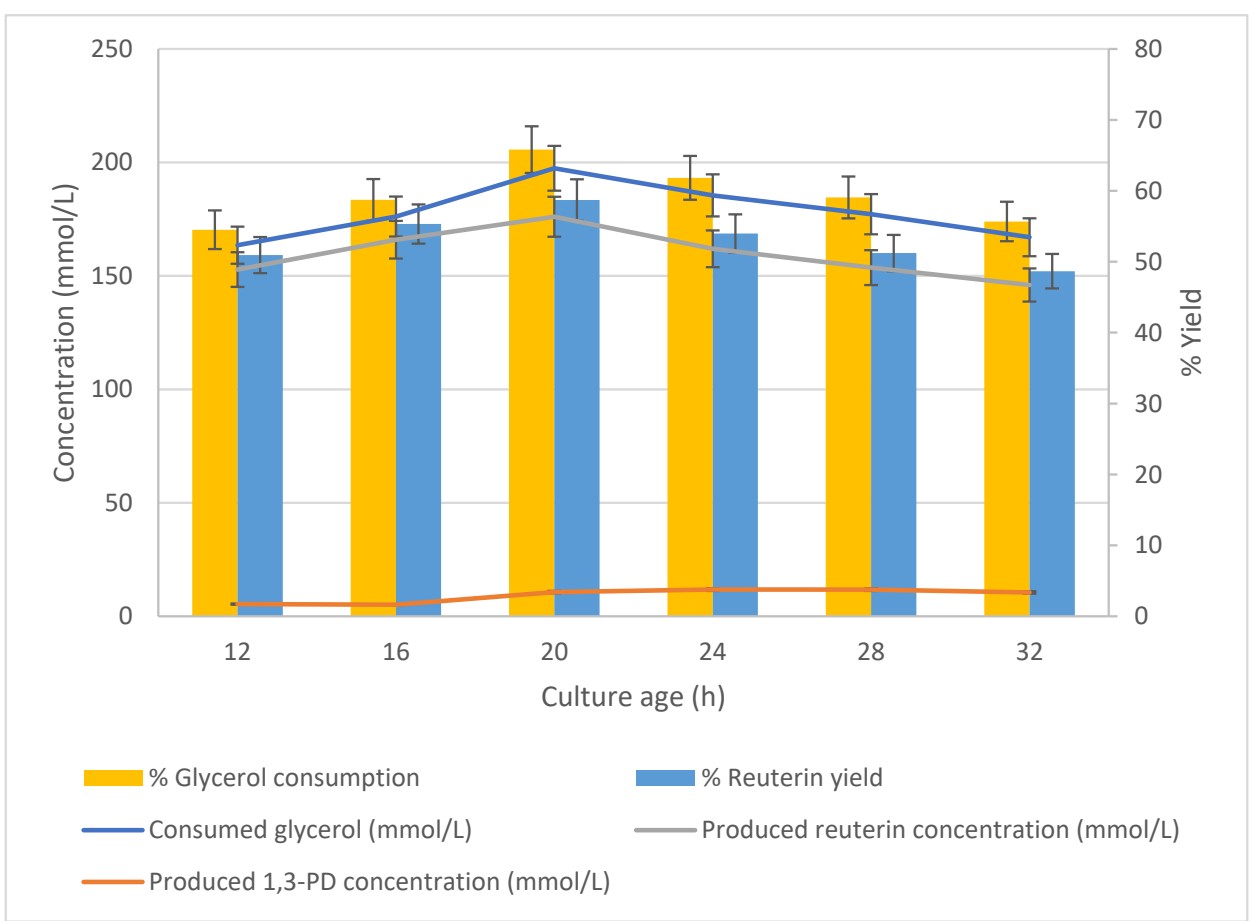

**Figure 9.** Effect of culture age on reuterin production.

### 3.7. The Rank Effect on Production of Reuterin for Six Single Factors

The Principal Component Analysis (PCA) method was used to determine the interrelationships between the six independent factors investigated in this study. The correlation ($p < 0.05$) matrices between the six factors and reuterin production are shown in Table 1. Results showed that biomass concentration, pH, glycerol concentration and temperature were positively correlated with reuterin production. Meanwhile incubation time and age of the culture showed a negative correlation with reuterin production. The correlation coefficient between reuterin production and glycerol concentration was 0.612, and this was the main influencing factor (Table 1). The second most important factor was pH (0.212). The other factors were incubation time ($-0.131$), biomass concentration (0.113) and temperature (0.103). The correlation of $-0.072$ for culture age and incubation time was the least influential factor for reuterin production.

PCA method was used to determine the relationships between biomass concentration, pH, glycerol concentration, temperature, incubation time and culture age and reuterin production; $n = 3$; analyses of experiments were triplicated.

Overall, the ranked effect of the six single factors on the production of reuterin was glycerol concentration > pH > incubation time > biomass concentration > temperature > culture age. Wan [18] reported slightly different results for the ranked effects of the production of reuterin, which were glycerol > biomass > incubation time > temperature > pH. The differences in rank for pH and biomass concentration between the previous study [18] and the current work were probably caused by differences in the strains used.

**Table 1.** Correlation Matrix ($p < 0.05$) for factors affecting reuterin production.

| Factor | Biomass, g/L | pH | Initial Glycerol, mmol/L | Temperature, °C | Incubation Time, h | Culture Age, h | Reuterin Production, mmol/L |
|---|---|---|---|---|---|---|---|
| Biomass, g/L | 1.000 | 0.096 | 0.009 | −0.069 | 0.079 | −0.034 | 0.113 |
| pH | 0.096 | 1.000 | −0.011 | 0.083 | −0.096 | 0.041 | 0.212 |
| Initial Glycerol mmol/L | 0.009 | −0.011 | 1.000 | 0.008 | −0.009 | 0.004 | 0.612 |
| Temperature, °C | −0.069 | 0.083 | 0.008 | 1.000 | 0.069 | −0.029 | 0.103 |
| Incubation Time, h | 0.079 | −0.096 | −0.009 | 0.069 | 1.000 | 0.034 | −0.131 |
| Culture Age, h | −0.034 | 0.041 | 0.004 | −0.029 | 0.034 | 1.000 | −0.072 |
| Reuterin Production, mmol/L | 0.113 | 0.212 | 0.612 | 0.103 | −0.131 | −0.072 | 1.000 |

## 4. Discussion

It was expected that reuterin production would be directly proportional to the biomass concentration. However, the optimum biomass concentration was 21 g/L. Reuterin production decreased at lower or higher biomasses. This effect was probably caused by the extent of surface area contact between the cells and the glycerol solution [18]. At lower concentrations of cells, the bacteria were sufficiently immersed in the glycerol solution, which enhanced reuterin production. As the biomass increased, the contact area between the cells and the glycerol solution decreased, probably due to cell aggregation. During the experiment, it was observed that the cells tended to sediment and adhere to the bottom and the walls of the bottles. Thus, this adhesion tendency might contribute to the lower reuterin production at higher concentrations of DPC16 biomass. Hence, reuterin production could possibly be increased by stirring the cell suspension, as it would increase the contact area of the DPC16 cells. This hypothesis needs to be verified in future experiments. In industrial production, to determine the optimized production conditions, it is important to balance the biomass concentration and glycerol concentration.

The results of this study were similar to those reported using *Limosilactobacillus reuteri* ATCC53608 [14] and *Limosilactobacillus reuteri* CG001 [24]. However, in the latter, it was reported that reuterin production continued to increase as the biomass concentration increased. In that study, the yield of reuterin reached 97.9% at 25.3 g/L of biomass used.

Different *L. reuteri* strains are reported to have different optimum pH levels to convert glycerol to reuterin, due to differences in the enzyme glycerol dehydratase. Generally, the conversion of glycerol to reuterin prefers a neutral to an acidic environment (pH 6.2 to 7.2). Wan and Wu [18] reported that reuterin production of *L. reuteri* ATCC53608 cells was 5% lower in glycerol solution without a phosphate buffer compared to glycerol solution adjusted using a buffer. The explanation was that *L. reuteri* produced short-chain fatty acids during its metabolism [25], which can lower the pH of the glycerol solution. The more acidic environment (pH < 6.2) can reduce the reuterin production ability of DPC16. *L. reuteri* DPC16 prefers a neutral environment for growth (pH 6.8 to 7.2), which also matches the pH in the intestinal tract (pH 6.8 to 7.2). Reuterin production in the present work was at its maximum in the range pH 6.2 to 7.2. This result was similar to previous reports [15,24]. However, Wan and Wu [18] reported that the optimum pH for reuterin production by *L. reuteri* ATCC53608 was 6.2, while Yao et al. [26] reported that the optimum initial pH for *L. reuteri* IMAU10240 of pH 6.5.

Glycerol is the substrate for reuterin production, and its concentration significantly affects reuterin production and yields [18]. The present results showed that there was an optimum concentration of glycerol, above which its conversion and reuterin production declined. The most likely reason for this outcome is that above a certain level, reuterin becomes toxic to the cells [26], and once a certain concentration is reached, production ceases. A possible solution to this problem is to use a technique for the continuous removal

of reuterin as it is formed. Similar results for initial glycerol concentrations that optimized the reuterin yield have been reported (<300 mmol/L using *L. reuteri* CG001 and *L. reuteri* ATCC53608) [18,24]. Thus, the initial glycerol concentration should be increased to no more than 350 mmol/L for effective reuterin production.

Temperature can affect reuterin production through two mechanisms. First, it is expected that as the temperature rises, the reaction rate will increase. However, conversely, as the temperature rises, enzymes become denatured and inactivated. The present results showed that, using resting cells, the optimum temperature for the conversion of glycerol to reuterin was 25 °C, which is considerably lower than the optimum growth temperature for this organism (37 °C). This illustrates the importance of using a two-step process, whereby organism growth can be optimized in the first step, and reuterin production in the second step. The observed result was similar to that reported by Doleyres [27], who showed that a relatively lower temperature was beneficial to reuterin production. However, others have drawn a different conclusion, reporting that relatively higher temperatures (30 °C and 37 °C) promoted glycerol-to-reuterin conversion [15,24].

The present results showed that under the experimental conditions used, the maximum production of reuterin occurred after 2 h of incubation, after which its concentration decreased. This can be partly explained by the further conversion of reuterin to 1,3-propanediol, but this does not explain all the losses. It is possible that glycerol was also converted into dihydroxyacetone, producing $NADH_2$. This extra $NADH_2$ could also be used in the conversion of reuterin to 1,3-PD. In addition, the toxic effect of reuterin on the cells must also be considered.

An additional factor to consider is the lack of additional energy sources for the cells. It is possible that glycerol's conversion to reuterin may be maintained by providing a source of energy, such as glucose. This could be tested in future experiments. Lüthi-Peng et al. [15] reported that the optimum incubation time for *L. reuteri* ATCC53608 was 3 h, while Wan and Wu [18] reported an optimum incubation time of only 1 h.

Finally, the present results showed that the culture age at harvest from the growth stage had only a minor effect on reuterin production and yields. However, it is apparent that harvesting the cells in the early stationary phase of growth is the most suitable when using resting cells in a two-step process. A previous report showed a similar result, with the optimum culture age for *L. reuteri* CG001 being 16 h to 24 h [24]. However, an optimum culture age for *L. reuteri* ATCC53608 was reported to be 8 h [15], while Wan and Wu [18] reported an optimum age in the early stationary phase (26 h).

## 5. Conclusions

In this work, the conversion of glycerol to produce reuterin in a two-stage process using harvested cells of *L. reuteri* DPC16 was studied. The conversion was affected by the culture age, glycerol concentration, pH, temperature, incubation time and biomass concentration. The glycerol concentration was the main influencing factor. The highest reuterin yield observed was 61.3% using 24-h precultured cells at a concentration of 21 g/L to convert 300 mmol/L of initial glycerol solution (pH 6.2) in 1 h at 37 °C. Cells harvested after approximately 20 h of growth were the most useful for glycerol-to-reuterin transformation at 25 °C and pH 6.8. The optimum biomass concentration was 25 g/L after incubation for 2 h.

**Author Contributions:** Conceptualization: N.G.-M. and Q.S.; investigation, methodology, formal analysis, data curation, Y.S.; writing–original draft: Y.S.; writing–review & editing: Y.S., N.G.-M., A.N.M. and I.S.M.; project administration: N.G.-M. All authors have read and agreed to the published version of the manuscript.

**Funding:** This research received no external funding.

**Institutional Review Board Statement:** Not applicable.

**Informed Consent Statement:** Not applicable.

**Data Availability Statement:** Not applicable.

**Acknowledgments:** We thank the Qingdao Institute of Bioenergy and Bioprocess Technology, the Chinese Academy of Science and the Qingdao Agricultural University for assistance with analyses.

**Conflicts of Interest:** The authors declare no conflict of interest.

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
