# Peer review of "Influence of Operating Conditions on Reuterin Production Using Resting Cells of Limosilactobacillus reuteri DPC16"

_fermentation, doi:10.3390/fermentation8050227_

Round 1

Reviewer 1 Report

The author confirmed the effect of biomass concentration, pH, glycerol concentration, temperature, incubation time, and culture age on ruterin conversion. Each experiment was an independent experiment to confirm which conditions affect ruterin production or yield. In addition, the importance of each factor was confirmed through Principal Component Analysis. The content of the manuscript is fitting to the advancement of the field, however cannot be published in its present version according to the following comments:

  • Lines 15-17: The author has not conducted experiments that satisfy the above conditions. Individual experiments were conducted under each condition, but the contents presented in the abstract do not match the results. This is an ambiguous statement and can be misleading.
  • Lines 138-146 are the result. It should be written in the results. Also add discussion about the Principal Component Analysis results.
  • Equation 3 should be corrected as follows: Reuterin yield (%) = reuterin concentration (mmol/L) x 100 / initial glycerol concentration (mmol/L)
  • What does the error bar in Figures 2-7 indicate? Generally, the average value is located in the center of the error bar. The Figures are difficult to understand because of the improper positioning of the error bar.
  • Since the % Glycerol consumption and % Reuterin yield units in the figure are calculated in %, they should be expressed as a percentage. The right y-axis in Figures 2-7 should be corrected.
  • How was dry cell mass calculated? If it was calculated using optical density, the calculation equation must be written down.
  • In general, results should be written in sections. Also, it's a good idea to add a title for your results.
  • While ruterin yield is also important, it is difficult to know how much ruterin is produced. Add a numerical indication of ruterin production concentration to the each results.
  • line 8 Lactobacillus reuteri Limosilactobacillus reuteri needs to be corrected.
  • line 160 37°C ) needs to be corrected.

Author Response

Dear Reviewer

Regards

Arthur

Reviewer 2 Report

I have reviewed the manuscript Fermentation--1716052 “Influence of Operating Conditions on Reuterin Production using Resting Cells of Limosilactobacillus reuteri DPC16”.

Comments: This work presents important results involving the use of resting cells to produce reuterin from glycerol. However, some corrections are necessary.

Abstract:

- Line : change “fermenting” to “incubating”;

Introduction:

- I suggest inserting the importance of converting glycerol, a by-product of biodiesel production, into biotechnological products;

- It is necessary to better justify the use of resting cells as biocatalysts in bioprocesses;

- Lines 65-66: show the reactions involved in a figure;

Experimental

- Line 122: Correct to “The analysis of acrolein concentration…”;

- Lines 139-151: These two paragraphs consist of the description of results and not methodology;

Results and discussion

- Lines 155-157: This is methodology and not results;

- Lines 270, 278, 312, 319: correct to “Wan and Xu”;

- Line 279: correct to “Yao et al.”;

- Line 311: correct to “Luthi-Peng et al.”;

- I suggest that the authors cite and compare the work of Jackson et al. (DOI:10.1002/mbo3.926), who evaluated resting cells to convert glycerol into glyceric acid and dihydroxyacetone;

- I suggest citing more works involving resting cells in biocatalysis.

Author Response

Dear Reviewer

Regards

Arthur

Round 2

Reviewer 1 Report

All requested have been properly revised.

I asked for correction of error bar and right y-axis in Figures 2-7.

Author response replied that it was modified, but I couldn't confirm it.

If this problem is resolved, It is possible to publish in Fermentation without further revision.